# Associations of Physical Activity and Handgrip Strength with Health-Related Quality of Life in Older Korean Cancer Survivors

**DOI:** 10.3390/cancers14246067

**Published:** 2022-12-09

**Authors:** Jeonghyeon Kim, Seamon Kang, Donghyun Kim, Hyunsik Kang

**Affiliations:** 1College of Sport Science, Sungkyunkwan University, Suwon 16419, Republic of Korea; 2Department of Sports and Health Science, Hanbat National University, Daejeon 34158, Republic of Korea

**Keywords:** cancer survivors, health-related quality of life, physical activity, handgrip strength

## Abstract

**Simple Summary:**

Along with a complete cure for cancer, health-related quality of life (HRQoL) is equally important to many cancer patients and survivors. Uncertainty exists regarding the associations of physical activity (PA) and relative handgrip strength (RHGS) with health-related quality of life (HRQoL) among elderly Korean cancer survivors. The current findings of the study support the prognostic roles of PA and RHGS in determining the HRQoL of older Korean cancer survivors. This study is unique to report that the negative influence of physical inactivity on HRQoL was observed in the cancer survivors of weak RHGS, but not in the cancer survivors of normal RHGS.

**Abstract:**

Background: Uncertainty exists regarding the associations between physical activity (PA), relative handgrip strength (RHGS), and health-related quality of life (HRQoL) among elderly Korean cancer survivors. This study investigated the moderating effect of RHGS on the relationship between PA and HRQoL in 308 cancer survivors aged ≥65 years using the data from the Korean National Health and Nutrition Examination Survey in 2014–2019. Methods: HRQoL was evaluated with the EuroQol-5-dimension instrument. PA and handgrip strength were assessed with the Global Physical Activity Questionnaire and a hand dynamometer, respectively. Age, sex, body mass index, income, marital status, educational background, heavy alcohol consumption, smoking status, multimorbidity, and hemoglobin were included as covariates. Results: Bivariate logistic regression showed that insufficient PA and no PA were associated with higher odds of a low HRQoL (odds ratio, OR = 2.6, 95% confidence interval, CI = 1.3~5.1, *p* = 0.005; OR = 2.4, 95% CI = 1.1~5.0, *p* = 0.024, respectively), compared with sufficient PA (OR = 1). Weak RHGS was associated with high odds of a low HRQoL (OR = 2.6, 95%= 1.530~4.6, *p* < 0.001) compared with normal RHGS (OR = 1). Particularly, RHGS (β = −0.0573, 95% CI = −0.1033~−0.0112) had a significant moderating effect on the relationship between PA and HRQoL even after adjustments for all the covariates. The negative influence of physical inactivity on HRQoL was observed in cancer survivors with weak RHGS but not in cancer survivors with normal RHGS. Conclusions: The current findings suggest that maintaining or promoting muscular strength through regular exercise is critical for the HRQoL of elderly Korean cancer survivors.

## 1. Introduction

Due to early detection and ever-expanding treatment options, the overall cancer mortality rate continues to decrease, and the number of cancer survivors continues to increase [1]. Despite improvements in diagnosis and treatment, however, a complete cure for cancer is the hope for most patients [2]. Equally important to many cancer patients and survivors is health-related quality of life (HRQoL), which is a generally accepted, multidimensional concept to describe how disease and treatment affect a patient’s perceived physical and mental health [3].

From diagnosis to treatment, cancer patients and survivors experience a wide range of symptoms and side effects, such as weakness, fatigue, and declines in physical functioning [4,5], which negatively influence HRQoL. Likewise, geriatric cancer patients and survivors experience persistent symptoms, such as discomfort, fatigue, weakness, activity restriction, depression, suicidal ideation, and comorbidities, which impair HRQoL [4,6]. Cancer and its treatment facilitate the aging process, tax cellular inflammation processes, inflict harm on major organ systems, and increase the risk of chronic diseases [7].

Physical inactivity in cancer survivors can cause muscle weakness, declines in physical functioning, delayed full recovery from treatment, and delayed return to normal life [8]. Sarcopenia, which is defined as a progressive loss of muscle mass or function with normal aging, is especially prevalent among cancer patients and survivors [9]. Sarcopenia increases the toxicity of anticancer treatment, decreases response to anticancer treatment and survival, and worsens with anticancer treatment [10]. It is generally considered a poor prognostic factor in cancer treatment and outcomes [11] and causes physical disability and decreased quality of life [12].

By contrast, physical activity ameliorates many of the problems experienced by cancer patients and survivors [13]. Physical activity is also associated with reduced odds of developing sarcopenia and increased odds of reversing sarcopenia in cancer patients [10]. Substantial evidence from randomized trials supports the potential for physical activity to reduce cancer-related symptoms and improve physical functioning [14,15,16] and HRQoL [17,18]. Considering the therapeutic potential of physical activity and muscular strength against the symptoms of cancer and the side effects of anticancer treatments, physical activity is recommended highly before, during, and after treatment [19].

Physical activity and muscular strength are two key lifestyle elements that favorably affect HRQoL [20,21], and they are interrelated to the point where some people with regular physical activity may also have adequate muscle strength and vice versa, suggesting the importance of taking both into account when assessing HRQoL. However, it is unclear how both physical activity and muscular strength affect the HRQoL of South Korean cancer patients and survivors. In this study, we investigate whether muscular strength moderates the effect of physical inactivity on HRQoL among Korean cancer survivors aged 65 years and older.

## 2. Materials and Methods

### 2.1. Data Source

We used data from the 2014–2019 period extracted from the sixth and seventh editions of the Korean National Health and Nutrition Examination Survey (KNHNES VI-VIII), a nationwide survey examining the health status, health behaviors, and food and nutrient consumption of the Korean population. Figure 1 shows the selection procedure for study participants. Geriatric cancer survivors were defined as living adults aged ≥65 years with a history of cancer of any type [21]. In brief, 37,491 adults aged ≥19 years participated in the KNHNES VI-VIII surveys. For our study purpose, we included only 10,484 adults aged ≥65 years. Subjects with no cancer (*n* = 6758) and subjects missing information about handgrip strength (*n* = 1557), health behaviors (*n* = 741), HRQoL (*n* = 457), or our covariates (*n* = 837) were excluded. The remaining 308 geriatric cancer survivors (106 males/34.4%) were included in the final analyses. Detailed information regarding KNHNES is available through the national public database (https://knhanes.kdca.go.kr/knhanes/sub04/sub04_04_01.do, accessed on 11 July 2022).

### 2.2. Measurements

#### 2.2.1. Health-Related Quality of Life

HRQoL was assessed using EuroQol-5 dimension (EQ-5D) in a Korean version [22]. The EQ-5D instrument measures the quality of life in the five dimensions of mobility, self-care, usual activities, pain/discomfort, and anxiety/depression, with each dimension rated on three levels (no problems, some or moderate problems, or extreme problems). The EQ-5D index was calculated using the Korean valuation set [23], and the lowest quartile of the index was deemed a low HRQoL.

#### 2.2.2. Physical Activity

Physical activity data were obtained using a Korean version of the Global Physical Activity Questionnaire (K-GPAQ). The K-GPAQ was used to assess walking (3.3 METs), moderate-intensity (4.0 METs), and vigorous-intensity (8 METs) activities (i.e., at work, transportation, exercise, and recreation) that lasted for at least 10 min per day during the past 7 days. The total volume of weekly physical activity (MET-minutes/week) was calculated based on intensity (METs), duration (minutes per day), and frequency (days per week). Physical activity was categorized as sufficient (≥600 METs-minutes/week), insufficient (1~<600 METs-minutes/week), or none (0 MET-minutes/week) according to the global recommendation for physical activity [https://www.who.int/news-room/fact-sheets/detail/physical-activity, accessed 10 July 2022]. The validity and reliability of the GPAQ were previously confirmed in the Korean population [24].

#### 2.2.3. Handgrip Strength

Handgrip strength was measured using a digital hand dynamometer (T.K.K 5401, Takei Scientific Instruments Co., Ltd., Tokyo, Japan). While in a standing position, each person exerted their maximal handgrip strength three times with each hand, with a 30-s rest between measurements [25]. To minimize variation in maximum handgrip strength between the dominant and non-dominant hands, we averaged the three values from the right and left hands and express the result as a relative term (kg/body mass index) [25]. For this study, we categorized the lowest quartile of relative handgrip strength as weak.

#### 2.2.4. Covariates

The covariates measured in this study were age (years), sex (male vs. female), body mass index (BMI), income (Korean won per month), marriage (married with a spouse vs. married without a spouse vs. never married), education (elementary or less vs. middle or high school vs. college or higher), heavy alcohol consumption, smoking status, multimorbidity, and hemoglobin [25,26]. Covariates were assessed using a self-reported questionnaire [26]. Past (at least 100 cigarettes in their lifetime) and current smokers were categorized as smokers, and the others were categorized as non-smokers. Heavy alcohol drinkers were defined as those who consumed > 14 drinks per week for males and >7 drinks per week for females [27]. Multimorbidity was defined as being diagnosed with two or more chronic conditions [28]. Serum hemoglobin was measured based on the SLS hemoglobin detection method with a Sysmex XN-9000 Automated Hematology Analyzer (Sysmex Corporation, Kobe, Japan).

### 2.3. Statistics

The normality of data was checked using QQ-plots and histograms. All data are presented as means (standard deviations) and numbers of cases (percentages) for continuous and categorical variables, respectively. Student’s *t*-test and chi-square test were used to compare the means and percentages of continuous and categorical variables, respectively. Bivariate logistic regression was used to estimate the odds ratios (ORs) and 95% confidence intervals (CIs) of a low HRQoL according to physical activity and handgrip strength. A linear regression analysis was used to estimate the correlation coefficients of the determinants for HRQoL. Finally, a moderation analysis of RHGS (moderator, W) in the relationship between physical activity (continuous, X) and HRQoL (continuous, Y) was conducted using the PROCESS macro by Andrew Hayes, as shown in Figure 2. Bias-corrected bootstrapping (*n* = 10,000) and 95% CIs were used to evaluate the statistical significance of the model. Otherwise, statistical significances were evaluated at α = 0.05 with SPSS-PC (version 27.0, IBM Corporation, Armonk, NY, USA).

## 3. Results

Sex differences in cancer epidemiology are one of the most significant findings from previous studies [29]. In this perspective, Table 1 represents the descriptive statistics of the study participants by sex. Older male cancer survivors were better educated (*p* < 0.001), more likely to live with a spouse (*p* < 0.001), more likely to be smokers (*p* < 0.001), and more likely to have heavy alcohol consumption (*p* < 0.001) than older female cancer survivors. Additionally, older male cancer survivors had a lower likelihood of multimorbidity (*p* < 0.001), fewer problems with mobility (*p* = 0.002) and pain or discomfort (*p* < 0.001), higher EQ-5D index scores (*p* = 0.001), higher hemoglobin (*p* < 0.001), and higher RHGS (*p* < 0.001) than older female cancer survivors.

Table 2 presents the correlation coefficients for the determinants of HRQoL in the study population. HRQoL correlated inversely with age (*p* < 0.001), being female (*p* = 0.001), living without a spouse (*p* = 0.007), and multimorbidity (*p* = 0.042) and positively with income (*p* < 0.001), education (*p* = 0.003), hemoglobin (*p* = 0.002), PA (*p* = 0.016), and RHGS (*p* < 0.001).

Table 3 presents the odds of a low HRQoL according to the levels of physical activity and RHGS. Insufficient PA and inactivity (no PA) were associated with increased odds of a low HRQoL (OR = 2.6, 95% CI = 1.3~5.1, *p* = 0.005; OR = 2.4, 95% CI = 1.1~5.0, *p* = 0.024, respectively) compared with sufficient PA (OR = 1). Those increased odds for a low HRQoL were no longer statistically significant when the data were controlled for age, sex, income, marriage, education, smoking, alcohol intake, multimorbidity, hemoglobin, and RHGS. Similarly, weak RHGS was associated with increased odds of a low HRQoL (OR = 2.6, 95% CI = 1.5~4.6, *p* < 0.001) compared with normal RHGS (OR = 1). However, those increased odds for a low HRQoL remained statistically significant (OR = 2.0, 95% CI = 1.1~3.6, *p* = 0.034) even after adjustments for age, sex, income, marriage, education, smoking, alcohol intake, multimorbidity, hemoglobin, and PA.

The relationship between PA (X) and HRQoL (Y) and RHGS (W) is shown in Table 4. The moderation analysis showed that the effect of physical inactivity on HRQoL differed significantly with the level of RHGS (β = −0.0543 and 95% CI = −0.1002~−0.0084). That interaction remained statistically significant (β = −0.0573 and 95% CI = −0.1033~−0.0112) even after adjustments for age, sex, income, marriage, education, smoking, alcohol intake, multimorbidity, and hemoglobin. The interaction was further investigated to better understand the moderation effect of RHGS on the relationship between PA and HRQoL. As shown in Figure 3, the negative influence of insufficient PA and inactivity on HRQoL was observed in cancer survivors with weak RHGS but not in those with normal RHGS.

## 4. Discussion

This cross-sectional study investigated the association between PA, RHGS, and HRQoL in a sample of 308 cancer survivors aged ≥ 65 years using data from the 2014–2019 KNHNES in South Korea. Our findings demonstrate a significant association between a lack of PA and weak RHGS and low HRQoL among older Korean cancer survivors. These findings add to those of previous studies by identifying the roles of both RHGS and PA in the HRQoL of older cancer survivors. This study is the first to demonstrate how RHGS moderates the effects of physical inactivity on HRQoL among older Korean cancer survivors. In cancer survivors with weak RHGS, a lack of PA had a significant negative effect on HRQoL but did not in cancer survivors with a normal RHGS.

The findings of this study are in line with those of other studies in confirming a relationship between muscular strength and overall health, including physical and emotional well-being. For instance, in a cross-sectional study of 224 cancer survivors in South Korea (151 breast and 73 colorectal cancers), Park et al. [29] demonstrated that moderate-to-vigorous PA had dose-dependent positive correlations with quality of life and physical functioning and negative correlations with fatigue and dyspnea. Kim and Kim [30] analyzed the data from 999 Korean cancer survivors aged ≥ 19 years and found that a lack of aerobic and strength exercise was associated with high stress and low HRQoL, as well as an increase in cardiometabolic risk factors such as elevated resting blood pressure, elevated triglycerides, and elevated high-sensitive C-reactive protein level.

Similar findings have been reported about the relationship between handgrip strength and HRQoL in Korean cancer patients and survivors. For instance, Paek and Choi [21] found that weak handgrip strength was linked to poor HRQoL among 1037 Korean cancer survivors from KNHNES VI and VII. Kim et al. [31] analyzed the data from 392 cancer survivors and found that low handgrip strength correlated with problems with self-care and usual activities among male cancer survivors and problems with mobility, daily activities, pain/discomfort, and anxiety/depression in female cancer survivors. Furthermore, handgrip strength has been linked to metabolic syndrome [32] and masticatory difficulty [33] among Korean cancer survivors.

The associations between physical activity and muscular strength and HRQoL has also been reported in other ethnic populations [10,18,20]. In a cross-sectional study of 191 Chinese pediatric cancer survivors (aged 9 to 16 years), for instance, Cheung et al. [34] demonstrated that higher PA was associated with better physical, emotional, social, and school functioning, and that greater handgrip strength was associated with better physical and emotional functioning. Furthermore, regular exercise improves handgrip strength in older adults and is a predictor of physical functioning and quality of life [35], suggesting that exercise and handgrip strength are related to each other. Taken together, the results of the present and earlier studies point to the possibility that PA and muscular strength can positively contribute to HRQoL among older cancer survivors by reducing symptoms and side effects and maintaining physical and mental functioning. For cancer patients and survivors who may struggle with a lack of motivation and sedentarism, a home-based intervention is advised as a more sustainable and adaptable approach [36,37].

Several explanations can be given for our findings of the present study. First, it is likely that the adverse health outcomes of physical inactivity contribute to low HRQoL in older Korean cancer survivors. In line with our hypothesis, it has been well established that physical inactivity results in declines in physical and mental functioning and increases the risks of chronic diseases, and cancer recurrence, which all negatively influence the HRQoL of cancer survivors [38,39]. Second, sarcopenia could explain the effect of weak RHGS on HRQoL. Sarcopenia is a major health concern among old people because it causes declines in physical function, disabilities, physical inactivity, and geriatric syndromes, all of which influence HRQoL [40,41]. Third, the effect of weak RHGS on the relationship between PA and HRQoL can be also explained by sarcopenia [42]. However, PA and muscular strength are associated: PA can increase muscle strength, or vice versa [43]. Therefore, it is unclear whether the effect of physical inactivity on low HRQoL in the weak RHGS category is attributable to the loss of muscle strength itself or to sarcopenia secondary to physical inactivity.

This study has several strengths. First, we identified that RHGS and PA were positively associated with HRQoL in older cancer survivors. Second, we showed that RHGS moderated the relationship between physical inactivity and a low HRQoL in older cancer survivors. Third, the data used in this study were derived from nationally representative, well-designed, systematic surveys in South Korea. This study also has limitations. First, the cross-sectional design of this study limits our ability to establish causal relationships between exposures and outcomes. Second, because our data were confined to the Korean population, the results might not be generalizable to other ethnic populations.

## 5. Conclusions

The results of this study suggest the clinical significance of having good muscular strength itself and suggest that muscular strength may counteract the negative effects of physical inactivity on the HRQoL of older Korean cancer survivors.

## Figures and Tables

**Figure 1 cancers-14-06067-f001:**
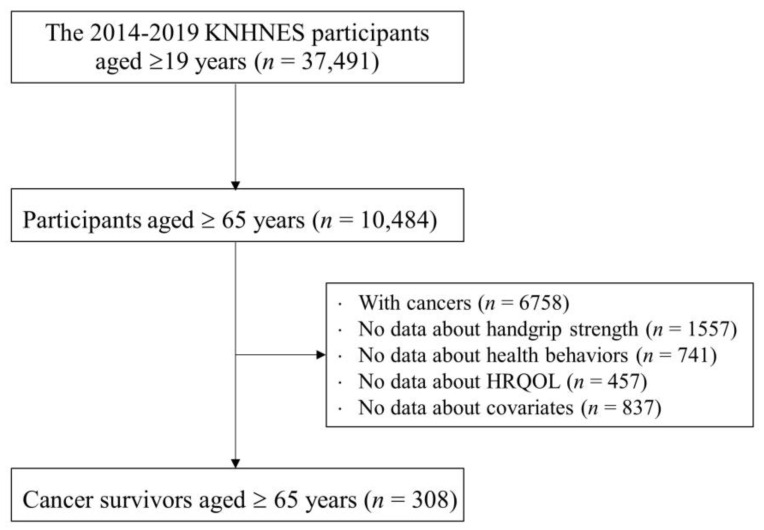
Flowchart for selection of study participants. KNHNES: Korean National Health and Nutritional Examination Survey; HRQOL: health-related quality of life.

**Figure 2 cancers-14-06067-f002:**
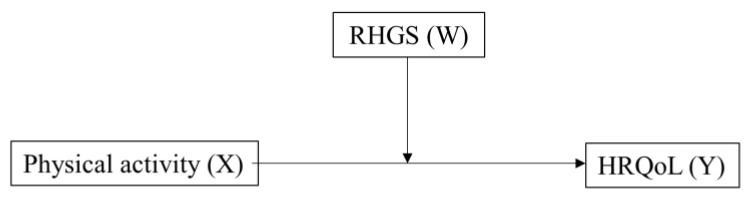
The relationship between physical activity (X) and health-related quality of life (HRQoL, Y) moderated by relative handgrip strength (RHGS, W).

**Figure 3 cancers-14-06067-f003:**
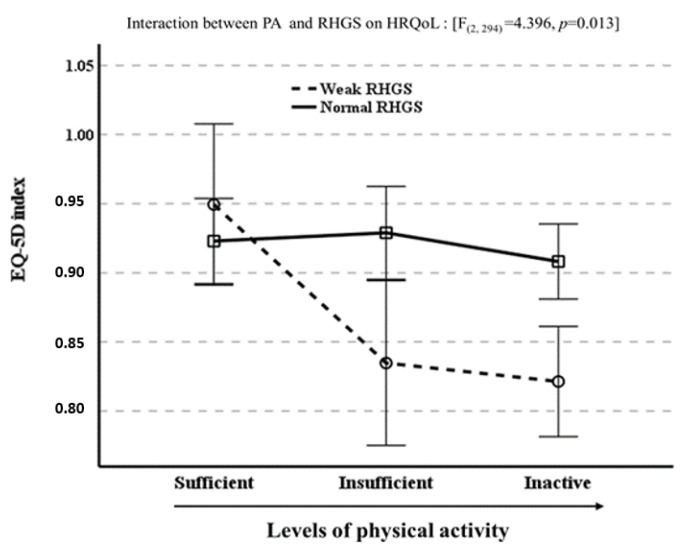
Moderating effect of relative handgrip strength (RHGS) on the relationship between physical activity (PA) and health-related quality of life (HRQoL). Physical activity was classified as sufficient (≥600 METs-minute/week) or insufficient (1~<600 METs-minutes/week) or inactive (sedentary or no physical activity).

**Table 1 cancers-14-06067-t001:** Descriptive statistics of study participants by gender.

Variables	Males (*n* = 202)	Females (*n* = 106)	Total (*n* = 308)	*p*-Value
Age (year)	72.4 ± 5.3	73.2 ± 5.2	72.9 ± 5.3	0.468
BMI (kg/m^2^)	23.1 ± 2.8	23.3 ± 3.1	23.2 ± 2.9	0.454
Income (10,000 won/month)	230 ± 231	192 ± 213	217 ± 226	0.156
Education, n (%)				<0.001
Elementary school or less	86 (42.6)	82 (77.4)	168 (54.5)	
Middle/high school	77 (38.1)	20 (18.9)	97 (31.5)	
College or higher	39 (19.3)	4 (3.8)	43 (14.0)	
Marital status, n (%)				<0.001
Married with a spouse	167 (82.7)	52 (49.1)	219 (71.1)	
Married without a spouse	32 (15.8)	53 (50.0)	85 (27.6)	
Never married	3 (1.5)	1 (0.9)	4 (1.3)	
Current/past smoking, n (%)	162 (80.2)	12 (11.3)	174 (56.5)	<0.001
Heavy alcohol, n (%)	21 (6.8)	0	21 (6.8)	<0.001
Multimorbidity, n (%)				<0.001
≥1	68 (33.7)	16 (15.1)	84 (27.3)	
>2	134 (66.3)	90 (84.9)	224 (72.7)	
EQ-5D problems				
Mobility, n (%)	57 (28.2)	48 (45.3)	105 (34.1)	0.003
Self-care, n (%)	10 (5.0)	9 (8.5)	19 (6.2)	0.220
Usual activities, n (%)	29 (14.4)	17 (16.0)	46 (14.9)	0.694
Pain/discomfort, n (%)	41 (20.3)	49 (46.2)	90 (29.2)	<0.001
Anxiety/depression, n (%)	20 (9.9)	17 (16.0)	37 (12.0)	0.116
EQ-5D index	0.92 ± 0.13	0.87 ± 0.15	0.90 ± 0.14	0.001
Hemoglobin (gm/dL)	14.1 ± 1.7	12.6 ± 1.4	13.6 ± 1.7	<0.001
PA (METs-minutes/week)	768 ± 1239	527 ± 1204	685 ± 1230	0.103
AHGS (kg)	30.3 ± 6.5	17.7 ± 4.4	26.0 ± 8.4	<0.001
RHGS (kg/BMI)	1.34 ± 0.30	0.77 ± 0.22	1.34 ± 0.38	<0.001

BMI: body mass index; EQ: EuroQol-5 dimension; PA: physical activity; RHGS: relative handgrip strength.

**Table 2 cancers-14-06067-t002:** Regression analyses for predictors of health-related quality of life.

Variables	β	SE	*p*-Value
Age	−0.213	0.001	<0.001
Body mass index	0.012	0.003	0.832
Sex	−0.184	0.016	0.001
Income	0.196	0.001	<0.001
Education	0.172	0.011	0.003
Marital status	−0.154	0.016	0.007
Smoking	0.080	0.016	0.164
Heavy alcohol	0.079	0.031	0.169
Multimorbidity	−0.116	0.018	0.042
Hemoglobin	0.174	0.004	0.002
Physical activity	0.137	0.001	0.016
Relative handgrip strength	0.297	0.020	<0.001

**Table 3 cancers-14-06067-t003:** Odds ratios (ORs) and 95% confidence intervals (CIs) of low health-related quality of life according to levels of physical activity and relative handgrip strength.

Predictors	Model 1		Model 2	
	Crude OR (95% CI)	*p*-Value	Adjusted OR (95% CI)	*p*-Value
Physical activity				
Sufficient	1 (reference)	0.004	1 (reference)	
Insufficient	2.6 (1.3~5.1)	0.005	1.5 (0.7~3.5)	0.311
Inactive	2.4 (1.1~5.0)	0.024	1.9 (0.9~4.0)	0.095
Relative handgrip strength				
Normal	1 (reference)		1 (reference)	
Weak	2.6 (1.5~4.6)	<0.001	2.0 (1.1~3.6)	0.034

Model 1: unadjusted. Model 2: adjusted for age, gender, income, marriage, education, smoking, alcohol intake, morbidity, hemoglobin, and physical activity (for handgrip strength) or handgrip strength (for physical activity).

**Table 4 cancers-14-06067-t004:** Moderation analysis of relative handgrip strength (RHGS) in the relationship between physical activity (PA) and health-related quality of life.

Predictors	Coefficients	SE	t	*p*	95% CI	
					Lower	Upper
Model 1 (R^2^ = 0.1334, F = 15.6028, *p* < 0.001)						
PA	0.1997	0.0474	4.2082	<0.001	0.1063	0.2930
RHGS	0.0916	0.0287	3.1596	0.0015	0.0352	0.1481
Interaction	−0.0543	0.0233	−2.3266	0.0206	−0.1002	−0.0084
R^2^ change due to the moderator = 0.0154 (F = 5.4129, *p* = 0.0206)						
Model 2 (R^2^ = 0.1732, F = 5.6375, *p* < 0.001)						
RHGS	0.2013	0.0534	3.7735	0.0002	0.0963	0.3064
PA	0.0896	0.0287	3.1264	0.0019	0.0332	0.1461
Interaction	−0.0573	0.0234	−2.4478	0.0150	−0.1033	−0.0112
R^2^ change due to the moderator = 0.0167 (F = 5.9917, *p* = 0.0150)						

Model 1 unadjusted. Model 2 adjusted for age, sex, income, marriage, education, smoking, alcohol intake, morbidity, and hemoglobin. SE: standard error; CI: confidence interval.

## Data Availability

The datasets used and analyzed during this study are available from the corresponding author upon reasonable request.

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
