# Peer review of "Associations of Physical Activity and Handgrip Strength with Health-Related Quality of Life in Older Korean Cancer Survivors"

_cancers, 2022, doi:10.3390/cancers14246067_

Round 1

Reviewer 1 Report

The manuscript is well written and provides interesting results on the physical activity, muscular strength, and quality of life thread. Some adjustments are suggested to improve the chance to be cited in the future. I do recommend answering them all.  

 Introduction

The introduction provides elements to reach the manuscript rationale. However, the final link in the last paragraph can be improved. For instance, I don’t see the necessity of mentioning anemia here. Previously you have provided information to understand the force decline in this population. State you can highlight that aging added to cancer can drastically affect the quality of life. However, the literature lacks studies that associate physical activity and muscle strength in this context. Some inference about the studied population is also valid.

“Considering the therapeutic potential of physical activity and muscular strength against the symptoms of cancer and the side effects of anticancer treatments, physical activity is recommended highly before, during, and after treatment.” The citation is missing here.

- Some hypothesis is missing.

Methods

-Improve the quality of Figure 1. The resolution is poor.

2.2.1 - Was the applied instrument in English? If so, all patients understood the language? Was this instrument validated?

Results

Why the sample was separated by sex in Table 1? If this is a relevant outcome for the study, some mention should be included in the introduction.

Table 1 – insert the legend below the table, and not inside it.

Figure 3 – I may have missed something, but how was the physical activity level classified as sufficient, insufficient, or inactive?

Discussion

The discussion is good. However, I recommend inserting some sentences discussing how muscular strength and physical activity can be improved in a home-based setting. To this end, I strongly recommend citing these studies:

-Adams, S. C., Iyengar, N. M., Scott, J. M. & Jones, L. W. Exercise implementation in oncology: One size does not fit all. J. Clin. Oncol. 36, 925–926 (2018).

-Hardcastle, S. J. & Cohen, P. A. Effective physical activity promotion to survivors of cancer is likely to be home-based and to require oncologist participation. J. Clin. Oncol. 35, 3635–3637 (2017).

-Hardcastle, S. J. & Cohen, P. A. Reply to S.C. Adams et al., C. Lopez et al., and R.U. Newton et al.. J. Clin. Oncol. 36, 928–930 (2018).

-Lopez, C., Jones, J., Alibhai, S. M. H. & SantaMina, D. What Is the “Home” in home-based exercise? The need to define independent exercise for survivors of cancer. J. Clin. Oncol. 36, 926–927 (2018).

-Kraemer, M. B., Priolli, D. G., Reis, I. G. M., Pelosi, A. C., Garbuio, A. L. P., & Messias, L. H. D. (2022). Home-based, supervised, and mixed exercise intervention on functional capacity and quality of life of colorectal cancer patients: a meta-analysis. Scientific Reports, 12(1), 1-13.

Author Response

In our Response to Reviewer #1

We deeply appreciate the reviewers for their thoughtful comments. We did our best to address all the comments/critics point-by-point, which are highlighted in yellow color. Two references are newly added, and they are listed on the last page.

 Introduction

Q1) The introduction provides elements to reach the manuscript rationale. However, the final link in the last paragraph can be improved. For instance, I don’t see the necessity of mentioning anemia here. Previously you have provided information to understand the force decline in this population. State you can highlight that aging added to cancer can drastically affect the quality of life. However, the literature lacks studies that associate physical activity and muscle strength in this context. Some inference about the studied population is also valid.

ANS1) In response to the comments, we revised the statements as follows (we removed the statements about anemia and revised the necessity of the study):

“Physical activity and muscular strength are two key lifestyle elements that favorably affect HRQoL [20,21], and they are interrelated to the point where some people with regular physical activity may also have adequate muscle strength and vice versa, suggesting the importance of taking both into account when assessing HRQoL. However, it is unclear how both physical activity and muscular strength affect the HRQoL of South Korean cancer patients and survivors.

 “Considering the therapeutic potential of physical activity and muscular strength against the symptoms of cancer and the side effects of anticancer treatments, physical activity is recommended highly before, during, and after treatment.” The citation is missing here.

The following article is cited.

Misiąg, W.; Piszczyk, A.; Szymańska-Chabowska, A.; Chabowski, M. Physical Activity and Cancer Care-A Review. Cancers (Basel) 2022, 14, 4154.

Q2) Some hypothesis is missing.

ANS2) We hypothesized that muscular strength would moderate the effect of physical inactivity on HHQoL in this study population. In our response to the comment, the last statement in the Introduction is rewritten as follows;

In this study, we investigate whether muscular strength moderates the impact of physical inactivity on HRQoL among Korean cancer survivors aged 65 years and older.

Methods

Q3) Improve the quality of Figure 1. The resolution is poor.

ANS3) Figures 1, 2 &3 are provided in 600 DPI.

Q4) 2.2.1 - Was the applied instrument in English? If so, all patients understood the language? Was this instrument validated?

ANS4) The EQ-5D is available in 20 different languages. We used a Korean version.  HRQoL was assessed using EuroQol-5 dimension (EQ-5D) in a Korean version [22].

Results

Q5) Why the sample was separated by sex in Table 1? If this is a relevant outcome for the study, some mention should be included in the introduction.

ANS5) In our response to the comment, sex differences in cancer morbidity and mortality are one of the most significant findings from previous studies [29]. We believe that the description of the study participants by gender is appropriate.

Q6) Table 1 – insert the legend below the table, and not inside it.

ANS6) Thanks. The legend is moved below the table.

Q7) Figure 3 – I may have missed something, but how was the physical activity level classified as sufficient, insufficient, or inactive?

ANS7) Thanks. In our response to the comment, the following information is added to the Figure legend: “Physical activity was classified as sufficient (³600 METs-minute/week) or insufficient (1 ~ <600 METs-minutes/week) or inactive (sedentary or no physical activity).”

Discussion

Q9) The discussion is good. However, I recommend inserting some sentences discussing how muscular strength and physical activity can be improved in a home-based setting. To this end, I strongly recommend citing these studies:

ANS9) Thanks. In our response to the suggestion, we added the following statement with two references cited (lines 8-10, page 9).

For cancer patients and survivors who may struggle with a lack of motivation and sedentarism, a home-based intervention is advised as a more sustainable and adaptable approach [36,37].

List of Added References

  1. Adams, S.C.; Iyengar, N.M.; Scott, J.M.; Jones, L.W. Exercise Implementation in Oncology: One Size Does Not Fit All. J Clin Oncol. 2018, 36, 925-926.
  2. Kraemer, M.B.; Priolli, D.G.; Reis, I.G.M.; Pelosi, A.C.; Garbuio, A.L.P.; Messias, L.H.D. (2022). Home-based, supervised, and mixed exercise intervention on functional capacity and quality of life of colorectal cancer patients: a meta-analysis. Scientific Reports 2022, 12, 1-13.

Reviewer 2 Report

Abstract

“The PROCESS macro by Andrew Hayes”: Please remove this bit of information from the abstract, it is inappropriate here.

It should be mentioned in the abstract that the relationship between insufficient/low physical activity was abrogated after controlling for covariates.

Introduction

HRQoL is very important to people living with and beyond cancer, but in my experience, I doubt that any patient would say it is as important as a complete cure for cancer! Please re-word.

How are you differentiating between the terms ‘cancer patient’ and ‘cancer survivor’? It would be helpful to the reader to define these terms early on.

Mentioning anemia here seems random and out of context. I appreciate anemia can lead to symptoms such as fatigue but so can many other cancer-related side effects. Moreover, you have not provided a reference. I suggest removing this sentence and instead focusing on literature that has investigated the relationship between physical activity/hand grip strength and HRQoL in cancer survivors (see comment below).

It is unclear why this study is needed in the context of what is already known on the topic. Many studies have investigated the relationship between physical activity, hand grip strength, and quality of life in cancer survivors (for example: https://www.ncbi.nlm.nih.gov/pmc/articles/PMC5984808/;   https://bmjopen.bmj.com/content/9/9/e030938). Please discuss these studies and justify why your study is needed in light of existing knowledge.

Methods

Data source: Did you prospectively register the study aims and data analysis plan? If so, please provide the reference ID. If you did not, please clearly state that the study was not prospectively registered.

Data source: Were participants assessed at a single time point or were they followed up at multiple time points? I appreciate information on KNHNES is available online but it would be helpful to clarify this in the text.

HRQoL: Please provide more information on how the EQ-5D-5L was scored. What was the range of possible scores when all five dimensions were combined? For example, a single index value based on the EQ-5D-5L value set for England ranges from -0.594 (worst possible health) to 1.0 (best possible health).

Physical activity: Please provide a citation for the physical activity reference values (sufficient >600 MET-min/week and insufficient <600 MET-min/week). Moreover, please could you justify why you did not split physical activity status into quartiles like you did for HRQoL and hand grip strength?

Covariates: The covariates you choose to include in the model should be based on theory – the choice of some covariates seem tenuous at best. Please justify the choice of these covariates. For example, please provide evidence that marital status impacts the relationship between physical activity/hand grip strength and HRQoL in cancer survivors?

 Statistical analysis: Please state the reference categories used in the models for each of the variables (physical activity, hand grip strength, HRQoL).

Statistical analysis: Have you considered making the anonymous data and SPSS syntax available on a publicly accessible repository?

Results

Table 1. Do you have information on the type of cancer survivors had?

Table 3. Reduce the number of decimal places for the odds ratios. I suggest using one decimal place.

Discussion

The odds ratios in Table 3 have very wide confidence intervals, suggesting a high level of uncertainty (possibly due to a small sample size). Please discuss this as a limitation.

“…and premature death, which all negatively influence the HRQoL of cancer survivors”. Premature death doesn’t influence HRQoL because the person is no longer alive! Please re-word.

These are interesting findings are there is a real opportunity here to have an in-depth discussion on the independent benefits of physical activity and being strong. In other words, being strong at baseline may bring benefits independent of being active – I suggest reading this paper https://www.liebertpub.com/doi/abs/10.1089/rej.2018.2111 and drawing out this argument in your discussion because it may help explain your findings. It would also be useful to acknowledge that hand grip strength does not appear to appreciably change with traditional resistance training.

Conclusion

Your conclusion does not reflect the data. Your findings suggest that being strong is important for HRQoL. It does not suggest that changes in strength through diet/exercise are important – these are two different things, as discussed above. In fact, your data seem to suggest that being strong is more important that exercising! Please revise accordingly.

Author Response

In our Response to Reviewer #2

We deeply appreciate the reviewers for their thoughtful comments. We did our best to address all the comments/critics point-by-point, which are highlighted in yellow color. Two references are newly added, and they are listed on the last page.

Abstract

Q1) “The PROCESS macro by Andrew Hayes”: Please remove this bit of information from the abstract, it is inappropriate here. It should be mentioned in the abstract that the relationship between insufficient/low physical activity was abrogated after controlling for covariates.

ANS1) Thanks. we revised the statement as follows: “Particularly, RHGS (β=-0.0573, 95% CI=-0.1033 ~ ‑0.0112) had a significant moderating effect on the relationship between PA and HRQoL even after adjustments for all the covariates.”

Introduction

Q2) HRQoL is very important to people living with and beyond cancer, but in my experience, I doubt that any patient would say it is as important as a complete cure for cancer! Please re-word.

ANS2) In our response to the comment, it is restated as follows: “however, a complete cure for cancer is the hope for most patients [2]”

Q3) How are you differentiating between the terms ‘cancer patient’ and ‘cancer survivor’? It would be helpful to the reader to define these terms early on.

ANS3) Cancer survivors are defined according to previous studies: “Geriatric cancer survivors were defined as living adults aged ³65 years with a history of cancer of any type [21].”

Q4) Mentioning anemia here seems random and out of context. I appreciate anemia can lead to symptoms such as fatigue but so can many other cancer-related side effects. Moreover, you have not provided a reference. I suggest removing this sentence and instead focusing on literature that has investigated the relationship between physical activity/hand grip strength and HRQoL in cancer survivors (see comment below).

ANS4) Anemia is out of context, and it is removed and rewritten as follows:

“Physical activity and muscular strength are two key lifestyle elements that favorably affect HRQoL [20,21], and they are interrelated to the point where some people with regular physical activity may also have adequate muscle strength and vice versa, suggesting the importance of taking both into account when assessing HRQoL. However, it is unclear how both physical activity and muscular strength affect the HRQoL of South Korean cancer patients and survivors.”

Q5) It is unclear why this study is needed in the context of what is already known on the topic. Many studies have investigated the relationship between physical activity, hand grip strength, and quality of life in cancer survivors (for example: https://www.ncbi.nlm.nih.gov/pmc/articles/PMC5984808/;   https://bmjopen.bmj.com/content/9/9/e030938). Please discuss these studies and justify why your study is needed in light of existing knowledge.

Q6) In response to the comments, we revised the statements as follows:

“Physical activity and muscular strength are two key lifestyle elements that favorably affect HRQoL [20,21], and they are interrelated to the point where some people with regular physical activity may also have adequate muscle strength and vice versa, suggesting the importance of taking both into account when assessing HRQoL. However, it is unclear how both physical activity and muscular strength affect the HRQoL of South Korean cancer patients and survivors. In this study, we investigate whether muscular strength moderates the effect of physical inactivity on HRQoL among Korean cancer survivors aged 65 years and older.”

Methods

Q7) Data source: Did you prospectively register the study aims and data analysis plan? If so, please provide the reference ID. If you did not, please clearly state that the study was not prospectively registered.

ANS7) Thanks for the comments. This is not a registered clinical trial. We used the data obtained from the sixth and seventh editions of the Korean National Health and Nutrition Examination Survey (KNHNES VI-VIII). This statement is clearly provided in the Methods.

“We used data from 2014-2019 extracted from the sixth and seventh editions of the Korean National Health and Nutrition Examination Survey (KNHNES VI-VIII), a nationwide survey examining the health status, health behaviors, and food and nutrient consumption of the Korean population.”

Q8) Data source: Were participants assessed at a single time point or were they followed up at multiple time points? I appreciate information on KNHNES is available online but it would be helpful to clarify this in the text.

ANS8) Participants were assessed at a single time point. The KNHNES data are available as suggested in the Methods: “Detailed information regarding KNHNES is available through the national public database (https://knhanes.kdca.go.kr/knhanes/sub04/sub04_04_01.do).”

Q9) HRQoL: Please provide more information on how the EQ-5D-5L was scored. What was the range of possible scores when all five dimensions were combined? For example, a single index value based on the EQ-5D-5L value set for England ranges from -0.594 (worst possible health) to 1.0 (best possible health).

ANS9) The EQ-5D is available in 20 different languages. We used a Korean version. HRQoL was assessed using EuroQol-5 dimension (EQ-5D) in a Korean version [22]. The range of possible scores are the same as the English version.

Q10) Physical activity: Please provide a citation for the physical activity reference values (sufficient >600 MET-min/week and insufficient <600 MET-min/week). Moreover, please could you justify why you did not split physical activity status into quartiles like you did for HRQoL and hand grip strength?

ANS10) Thanks. In our response to the comment, physical activity was classified according to the WHO recommendations, which is 600 METs-minutes per week (equivalent of 150 minutes of moderate-intensity activity or 75 minutes of vigorous-intensity physical activity a week (https://www.who.int/news-room/fact-sheets/detail/physical-activity).

Q11) Covariates: The covariates you choose to include in the model should be based on theory – the choice of some covariates seem tenuous at best. Please justify the choice of these covariates. For example, please provide evidence that marital status impacts the relationship between physical activity/hand grip strength and HRQoL in cancer survivors?

ANS11) HRQoL is influenced by social factors, including age, gender, living in couple, level of education,occupational status, net income per household, and others (Kivits J, Erpelding ML, Guillemin F. Social determinants of health-related quality of life. Rev Epidemiol Sante Publique. 2013 Aug;61 Suppl 3:S189-94).

 Q12) Statistical analysis: Please state the reference categories used in the models for each of the variables (physical activity, hand grip strength, HRQoL).

ASN12) Thanks. We described the reference categories as follows”

“Finally, a moderation analysis of RHGS (moderator, W) in the relationship between physical activity (continuous, X) and HRQoL (continuous, Y) was conducted using the PROCESS macro by Andrew Hayes, as shown in Figure 2.”

Q13) Statistical analysis: Have you considered making the anonymous data and SPSS syntax available on a publicly accessible repository?

ANS13) Thanks for the comments. The data and SPSS syntax are accessible through the national public database (https://knhanes.kdca.go.kr/knhanes/sub04/sub04_04_01.do).

Results

Q14) Table 1. Do you have information on the type of cancer survivors had?

ASN14) Thanks for the comment. Unfortunately, the information is not available.

Q15) Table 3. Reduce the number of decimal places for the odds ratios. I suggest using one decimal place.

ANS15) Thanks. We reduced the number to one decimal place (in Abstract, Table 3, and context), as suggested.

Discussion

Q16) The odds ratios in Table 3 have very wide confidence intervals, suggesting a high level of uncertainty (possibly due to a small sample size). Please discuss this as a limitation.

ANS16) Thanks. However, we do not agree with the comment about the small sample size. We believe that the sample size was enough to test our primary hypothesis, especially considering geriatric cancer survivors.

Q17) “…and premature death, which all negatively influence the HRQoL of cancer survivors”. Premature death doesn’t influence HRQoL because the person is no longer alive! Please re-word.

ANS17) Thanks. We removed “premature death” from the statement: “In line with our hypothesis, it has been well established that physical inactivity results in declines in physical and mental functioning and increases the risks of chronic diseases, and cancer recurrence, which all negatively influence the HRQoL of cancer survivors [36,37).”

Q18) These are interesting findings are there is a real opportunity here to have an in-depth discussion on the independent benefits of physical activity and being strong. In other words, being strong at baseline may bring benefits independent of being active – I suggest reading this paper https://www.liebertpub.com/doi/abs/10.1089/rej.2018.2111 and drawing out this argument in your discussion because it may help explain your findings. It would also be useful to acknowledge that hand grip strength does not appear to appreciably change with traditional resistance training.

ANS18) Thanks for the comments. Although we agree with your comments, we do not want to exaggerate our conclusion due to the cross-sectional nature of the current study. Instead, our conclusion statement is revised as suggested below.

 Conclusion

Q19) Your conclusion does not reflect the data. Your findings suggest that being strong is important for HRQoL. It does not suggest that changes in strength through diet/exercise are important – these are two different things, as discussed above. In fact, your data seem to suggest that being strong is more important that exercising! Please revise accordingly.

ANS19) Thanks for the comment. In our response to the comment, we revised the conclusion as follows;

“The results of this study suggest the clinical significance of having good muscular strength itself and suggest that muscular strength may counteract the negative effects of physical inactivity on the HRQoL of older Korean cancer survivors.”

List of Added References

  1. Adams, S.C.; Iyengar, N.M.; Scott, J.M.; Jones, L.W. Exercise Implementation in Oncology: One Size Does Not Fit All. J Clin Oncol. 2018, 36, 925-926.
  2. Kraemer, M.B.; Priolli, D.G.; Reis, I.G.M.; Pelosi, A.C.; Garbuio, A.L.P.; Messias, L.H.D. (2022). Home-based, supervised, and mixed exercise intervention on functional capacity and quality of life of colorectal cancer patients: a meta-analysis. Scientific Reports 2022, 12, 1-13.

Round 2

Reviewer 1 Report

The authors responded to all my notes, and they need to be congratulated for that. In its current format, the study presents an acceptable rationale and its results corroborate the objective.